# Medicinal Plants *Galega officinalis* L. and Yacon Leaves as Potential Sources of Antidiabetic Drugs

**DOI:** 10.3390/antiox10091362

**Published:** 2021-08-26

**Authors:** Halyna Hachkova, Mariia Nagalievska, Zoriana Soliljak, Olena Kanyuka, Alicja Zofia Kucharska, Anna Sokół-Łętowska, Elena Belonovskaya, Vyacheslav Buko, Nataliia Sybirna

**Affiliations:** 1Department of Biochemistry, Faculty of the Biology, Ivan Franko National University of Lviv, Hrushevskyi St. 4, 79005 Lviv, Ukraine; mariia.nagalievska@lnu.edu.ua (M.N.); solilyak.zoryana@gmail.com (Z.S.); kanukaolena@gmail.com (O.K.); nataliya.sybirna@lnu.edu.ua (N.S.); 2Department of Fruit, Vegetable and Plant Nutraceutical Technology, Faculty of the Biotechnology and Food Science, Wrocław University of Environmental and Life Sciences, Chełmońskiego St. 37, 51-630 Wrocław, Poland; alicja.kucharska@upwr.edu.pl (A.Z.K.); anna.sokol-letowska@upwr.edu.pl (A.S.-Ł.); 3Division of Biochemical Pharmacology, Institute of Biochemistry of Biologically Active Compounds, National Academy of Sciences, BLK, 50, 230030 Grodno, Belarus; ms.belonovskaya@yandex.by (E.B.); vu.buko@tut.by (V.B.)

**Keywords:** diabetes mellitus, *Galega officinalis* L., *Smallanthus sonchifolius* Poepp. and Endl., in vitro antioxidant activity, hypoglycemic effect, pancreatic β-cell

## Abstract

Hypoglycemic and antioxidant properties of extracts of medicinal plants *Galega officinalis* L. (aboveground part) and yacon (*Smallanthus sonchifolius* Poepp. & Endl.) (leaves) as potential sources of biologically active substances with antidiabetic action have been studied. The pronounced hypoglycemic effect of *Galega officinalis* extract, devoid of alkaloids, at a dose of 600 mg/kg in experimental diabetes mellitus (DM) has been proven. The established effect is evidenced by a decrease in the concentration of glucose and glycosylated hemoglobin in the blood, increase glucose tolerance of cells, increase C-peptide and insulin content in the plasma of rats’ blood. The effective hypoglycemic effect of the extract in the studied pathology was confirmed by histological examination of the pancreas. The cytoprotective effect of the studied extract on pancreatic cells at a dose of 1200 mg/kg was experimentally confirmed. In the standard cut area, an increase was found in the number of Langerhans islets, their average area, diameter, volume, and a number of β-cells relative to these indicators in animals with diabetes. Comparative screening of the antioxidant properties of 30, 50, 70, and 96% water–ethanol extracts of yacon indicates the highest potential of 50% water-ethanol extract to block free radicals in in vitro model experiments. The non-alkaloid fraction of *Galega officinalis* extract showed moderate antioxidant activity and was inferior to yacon extract in its ability to neutralize reactive oxygen species (ROS) and bind metal ions of variable valence. The level of antioxidant potential of the studied extracts is due to differences in the quantitative content of compounds of phenolic nature in their compositions. The obtained data on the biological effects of *Galega officinalis* extract on the structural and functional state of β-cells of the pancreas and antioxidant properties of *Galega officinalis* and yacon extracts substantiate the prospects of using these plants to create antidiabetic medicines and functional foods based on them.

## 1. Introduction

According to modern views, oxidative stress (OS) induced by hyperglycemia is considered to be the main mechanism of β-cell damage, which accelerates the progress of diabetes mellitus (DM) and its complications [1,2]. Therefore, it is important to create antidiabetic drugs and functional foods that will effectively prevent not only hyperglycemia but also the development of oxidative stress. Taking this into account, the role of antioxidants is actively studied both in the treatment of type 1 diabetes and in the prevention of its complications [3,4].

Preliminary screening of the antioxidant activity of potential drugs and phytomedicines, in particular, involves the use of methods for the initial assessment of antioxidant and anti-radical properties of biologically active components in in vitro experiments. In vivo, it is impossible to adequately assess the primary antiradical and antioxidant effects of biologically active substances, as they may be mediated by complex biochemical processes occurring in a living organism and are not the primary molecular mechanisms of action of test substances [5].

Given the multifactorial pathogenesis of diabetes, it is appropriate and promising to use naturally balanced antidiabetic medicine in the complex therapy of this disease. These drugs should show a complex multidirectional impact on various links of disease pathogenesis, for example, reduces blood glucose and insulin resistance. Medicinal plants due to their rich chemical composition show a wide range of pharmacological properties. Phytomedicines combine well with antidiabetic drugs, enhancing their therapeutic result and causing fewer adverse side effects while prolonged usage. Scientific studies confirm the safety and effectiveness of drugs of natural origin [6,7,8].

Today, many medicinal plants are known to be used in alternative medicine in many countries around the world to treat both types of diabetes, but the list of official antidiabetic drugs based on them is insufficient.

A promising raw material for the creation of antidiabetic drugs of natural origin is *Galega officinalis* L. Its hypoglycemic potential was established in 1927, but the literature on the sugar lowering effect of herbs and seeds of this species is contradictory. It was previously thought that the hypoglycemic action is inherent in alkaloids, as indicated in several reports [9,10,11,12]. To date, many synthetic drugs of the biguanide class have been created which in terms of the chemical structure and pharmacological properties are analogous to the alkaloid from *Galega officinalis*-galegine. The known oral hypoglycemic drug metformin is a prototype of galegine. At the same time, scientific data are indicating that the non-alkaloid fraction of *Galega officinalis* extracts possesses the hypoglycemic effect [13,14]. The contradictory nature of the literature data on the glucose-lowering action of *Galega officinalis* necessitates a more detailed study of this issue.

Another plant with a pronounced hypoglycemic effect is yacon—(*Smallanthus sonchifolius* (Poepp. & Endl.) H. Robinson), which also has anti-inflammatory, antimicrobial, and antioxidant properties [15]. Yacon is a rich source of various biologically active substances and phenolic compounds, in particular [16]. Metabolites of phenolic nature are one of the most important groups of natural antioxidants. Therefore, this plant can be considered a valuable raw material for the production of natural antioxidants. According to the literature, phenolic compounds are mainly concentrated in yacon leaves [17].

However, the level of research and implementation of phytotherapeutic agents based on these medicinal plants in diabetology is currently insufficient.

The study aimed to investigate the biological effects of the non-alkaloid fraction of *Galega officinalis* extract on structural and functional state of islets of Langerhans β-cells of the pancreas of rats with diabetes induced by streptozotocin, and antioxidant activity of yacon and *Galega officinalis* extracts with the prospect of using these extracts as a basis for drugs for prevention and treatment of diabetes.

## 2. Materials and Methods

### 2.1. Plant Material

The aboveground part (leaves and stems) of *Galega officinalis* L. raw material was collected in Lviv (Ukraine, June 2013). The leaves of *Smallanthus sonchifolius* (Poepp. & Endl.) H. Robinson) were gathered in the Kyiv region (Ukraine). The identification of plants taxonomic was made with the help of Senior Research Fellow Mariia Skybitska the curator of medicinal plants’ collection of Ivan Franko National University of Lviv Botanical Garden.

### 2.2. Preparation of Galega officinalis Extract and Its Stabilization by Adding the PS (Pseudomonas *sp.*) Biocomplex

Leaves and stems of *Galega officinalis* were gathered at the period of flowering. Afterwards, it was dried out on air and fragmented into even pieces. The extract was made by infusion of plant material in 96% ethanol at a ratio of 1:5. Extraction was performed for 12 h at a temperature of 20–23 °C. Subsequently, the ethanol extract was evaporated (in a vacuum at 50–55 °C) using rotary evaporator Laborota 4001 (Heidolph, Schwabach, Germany). The procedure of evaporation was performed until obtaining dense residue extract. The crude extract percentage yield was estimated as 15–17%. To obtaining non-alkaloid fraction of ethanol extract, an equal amount of H_2_O and CHCl_3_ was added. The obtained mixture was carefully shacked and then centrifuged (10 min, 600 g). Obtained by such a scheme chloroform part of the ethanol extract was evaporated (in a vacuum at 40–55 °C) until obtaining a solid residue. The chloroform fraction estimated percentage yield was 3.3–5%. For the stabilization of chloroform fraction, biocomplex PS (biosynthesized by *Pseudomonas* sp. PS-17 surface-active products) was used. To the obtained solid residue of chloroform fraction of *Galega officinalis* extract, water and biocomplex PS were added at a concentration of 3.3 g/L [18]. All procedures were followed by shaking in Vortex (Biosan, Latvia) (Figure 1).

The presented method of extraction and stabilization of obtained chloroform fraction allows obtaining a stable water emulsion deprived of poisonous alkaloids.

### 2.3. Induction of Diabetes

Streptozotocin (STZ, Sigma, St. Louis, MO, USA) was used for creating a model of DM type 1. Induction of disease was fulfilled with an intraabdominal injection of STZ (0.055 g/kg body weight dissolved before use in 10 mM citrate buffer, pH 5.5) to animals after overnight fasting. To evaluate the development of DM 3 days after STZ injection the glucose level in whole blood was measured. A glucose concentration of 14 mmol/L and above indicated the development of the studied pathology in animals. The medicines that were used for the treatment of DM were administered on the 14th day after induction of diabetes.

### 2.4. Experimental Animals

For the experiments, three-month-old male Wistar rats (150 to 220 g) were used. The animals were kept in the animal house of Ivan Franko National University of Lviv. Each group of animals was housed in separate polypropylene cages. Animal housing conditions included the following parameters: twelve hours light/dark cycle, the controlled temperature of 25 ± 2 °C, 45–55% relative humidity, a standard laboratory diet, and water ad libitum. For rats feeding was used a standard full-fledged balanced granular chow (Rezon-1 (recipe K 120-1), Vita Poshtova, Ukraine), which provided the physiological needs of their body in vitamins, trace elements, and minerals. Beforehand starting experimental manipulation animals undergo 7 days acclimatization period to adapt to their environment. Rats were deprived of access to food twelve hours before and during the experiment. The health status of the rats was monitored systematically, and no adverse events were recorded during the housing period. The protocol was conducted in compliance with the general ethical principles of animal experiments following the “General Principles of Work on Animals”, approved by the 1st National Congress of Bioethics (Kyiv, Ukraine, 2001) and the Law of Ukraine No. 3447-IV “On Protection of Animals from Cruelty” from 26 February 2006, and conforming the guidelines from Directive 2010/63/EU of the European Parliament on the protection of animals used for scientific purposes (the Directive is firmly based on the principle of the Three Rs, to replace, reduce, and refine the use of animals used for scientific purposes), as well as approved by the Ethics Committee of Ivan Franko National University of Lviv, Ukraine (protocol No. 23-07-2021 from 19 July 2021). Rats were unsystematically divided into the groups (*n* = 5–8/group): control animals (C); control animals that were treated with extract of *Galega officinalis* at doses 600 mg and 1200 mg/kg per day (C + G) for 14 days; animals with diabetes mellitus (D); animals with diabetes that were treated with extract of *Galega officinalis* at doses 600 mg and 1200 mg/kg per day (D + G) starting from the 15th day after the induction of diabetes (subject to confirmation of the disease) for 14 days. The non-alkaloid fraction of *Galega officinalis* extract was administered to rats *per os* using a tube at doses 600 mg/kg (one dose per day) and 1200 mg/kg (two doses of 600 mg/kg per day) daily for 14 days. All obtained results were included in the data analysis. Animals from group C + G and D + G received stabilized water emulsion of chloroform fraction *Galega officinalis* extract through a tube; animals from group C and D received water in the same way and period of a day.

### 2.5. Determination of Blood Glucose Concentration

The glucose oxidase method was used for glucose concentration determination. For this purpose, analytical kit was used for the enzymatic determination of blood glucose (Philisit-Diagnostics, Dnipro, Ukraine).

### 2.6. Immuno-Enzymatic Analysis of Insulin, C-Peptide, and Tumor Necrosis Factor Alpha Contents

The contents of insulin, C-peptide in plasma, and TNF-α in rats’ pancreas were determined by immuno-enzymatic analysis using standard ELISA kits (Sigma, USA) according to the protocol of the manufacturer kit. The measurements were performed on an Epoch microplate spectrophotometer (BioTek, Winooski, VT, USA).

### 2.7. Glucose Tolerance Test Assay

The assessment of glucose tolerance was conducted in the mornings after the 18-h fasting. The blood for the determination of glucose concentration was taken from the rat-tail vein before and after the carbohydrate load. Based on the obtained glucose concentration spectra, the glycemic curves were constructed. Analysis of these curves allows to estimate the rate of glucose assimilation and evaluate the effects of the extract on the level of blood sugar (the zero points are on an empty stomach and after glucose taking every ten minutes in a range from 10 to 120 min). Using the trapezoid rule, integral index of the area under the glycemic curve (AUCglu) was calculated [19]. AUCglu is an analytical tool for understanding the total response to the glucose tolerance test.

### 2.8. Determination of Glycosylated Hemoglobin Content

The content of glycosylated hemoglobin (HbA1c) in erythrocytes was determined by the colorimetric method, which is based on acid hydrolysis of the ketamine bond in the presence of oxaloacetate. The reaction produces 5-oxymethylfurfural, which, interacting with 2-thiobarbituric acid (TBA) forms a colored complex, the intensity of which was determined using a spectrophotometer HeliosEpsilon (ThermoScientific, Waltham, MA, USA) at a wavelength of 443 nm [20].

### 2.9. Light Microscopy of Cells of the Pancreas

For histological examination by light microscopy, pancreatic samples were fixed in Buena fluid for 12 h at +4 °C. After fixation, the material was dehydrated according to standard methods in ethanol with increasing concentration and poured into history. Histological sections 5 μm thick were stained with Mayer hematoxylin and eosin for light microscopy and morphometry [21], and chromium-hematoxylin staining according to Gomori [22] and aldehyde-fuchsin staining according to Gabe [23] were used to verify β-cells.

Evaluation of micro preparations, morphometry, and microphotography was performed using an Olympus CX-41 light microscope additionally equipped with digital camera (Olympus, Tokyo, Japan).

Morphometric analysis of the islet apparatus of the pancreas was performed using the computer program ImageJ (NIH, Bethesda, MD, USA). The density of the location of the islands (the number of islets of Langerhans on a standard area (N/10 mm^2^)), their individual area (S, μm^2^), diameter (Di, μm), volume (Vi, μm^3^), the number of β-cells per 1000 μm² were determined. Calibration was performed using an object micrometer, which found that the micrometers matched the pixels on the monitor screen.

The area of the islands was determined by the method of delineation. To calculate the diameter of the islands, largest (a) and smallest (b) axes were measured and calculated using the formula [24]:(1)$$D=\sqrt{ab}$$

Given that the islets of the pancreas are a spheroidal structure, to determine the average diameter of the islets was used the Fullman formula [24]:(2)$$D=\frac{\pi }{2}\times \;\frac{N}{\begin{array}{c} \frac{1}{d1}+\frac{1}{d2}\ldots \ldots \frac{1}{dN} \end{array}},$$
where *N* is the total number of measurements. This diameter was used to determine the average volume of the islets of the pancreas using the formula [24]:(3)$$V=\frac{4\pi }{3}\times {\left(\frac{D}{2}\right)}^{3}$$

The number of β-cells was determined by the method of spot counting after staining of histological sections with chromium-hematoxylin according to Gomori.

### 2.10. Preparation of Yacon Extracts

For research, we used the aboveground part (leaves), which were collected in the fall. From 1 kg of raw leaves, the yield of air-dry mass was 300 g.

From powdered yacon leaves water, ethanol, and water-ethanol (with an ethanol content of 30%, 50%, 70%, and 96%) extracts (WEE) were prepared by infusion of raw materials (ratio 1:10) for 5 h at 20 and 40 °C and pH 2–2.5. The obtained extracts were filtered and evaporated at 45–60 °C in a vacuum using a LABOROTA 4001 rotary evaporator (Heidolph, Germany).

### 2.11. Total Phenolic Compound Content (TPC) Analysis

Follin–Ciocalteu method was used for the analysis of total phenolic content. Phenol compounds when interacting with a Follin–Ciocalteu reagent (containing sodium tungstate, sodium molybdate, lithium sulfate, phosphoric and hydrochloric acids) forms a green-blue colored compound that was analyzed spectrophotometrically (765 nm). Total phenolic content was expressed as gallic acid equivalent per 1 g of extract (mg GAE/g) [25].

### 2.12. DPPH Assay

DPPH (2,2-diphenyl-1-picrylhydrazyl) assay is based on the ability of antioxidants to reduce 2,2-diphenyl-1-picrylhydrazyl. The decreasing of absorbance measured at 517 nm can assess the reduction of DPPH working solution, which is purple. Discoloration of DPPH solution shows that the previously unpaired electron has been paired. DPPH assay is a convenient method for the measurement of antioxidants’ free-radical scavenging ability [26].

### 2.13. ABTS Assay

ABTS (2,2′-azine-bis acid (3-ethylenebenzothiazoline-6-sulfonic acid) assay is the method of evaluation of antioxidant activity by quantitative assessment of free-radical scavenging ability of compounds. The method is based on the ability of tested components to quench the stable ABTS^•+^. Absorbance was measured at wavelength *λ* = 734 nm [27].

### 2.14. FRAP Assay

FRAP assay is the ferric-reducing antioxidant power test. Method is based on the reduction of ferric 2,4,6-tris(2-pyridyl)-1,3,5-triazine [Fe(III)-TPTZ] at low pH. In quantitative analyses, ferrous sulfate was used as the standard for relating the absorbance (*λ*  = 593 nm) with the concentration [28].

### 2.15. Statistical Analysis of Results

All the experiments for the determination of total phenolics and antioxidant properties using DPPH, ABTS, and FRAP were conducted in triplicates. The values are expressed as the mean ± standard deviation (SD). Statistical differences between two alternative data sets were performed using a *t*-test (two-way analysis of variance). The difference was considered significant under *p* ≥ 0.95 (the level of significance *p* < 0.05).

## 3. Results

### 3.1. Hypoglycemic Effect of Galega officinalis Extract

Under the conditions of DM, a pronounced hypoglycemic effect of the non-alkaloid fraction of *Galega officinalis* extract was established. Administration of this extract to diabetic animals resulted in a 63% reduction in blood glucose concentration compared to diabetes. However, in the case of *Galega officinalis* extract introduction to control animals, the blood glucose content did not significantly change (Figure 2).

Glycosylated hemoglobin (HbA1c) is an integral indicator of carbohydrate metabolism compensation in diabetes conditions. The level of HbA1c correlates with the amount of glucose in the blood, and also reflects the risk of complications of this pathology. Against the background of hyperglycemia, which develops in rats with diabetes, an increase in the content of glycosylated hemoglobin by 66% relative to control was shown. *Galega officinalis* extract caused a decrease by 41% in HbA1c content in animals with the studied pathology (Figure 2).

The antihyperglycemic effect of *Galega officinalis* extract was evaluated by its ability to reduce glucose at the maximum development of hyperglycemia after glucose loading. To evaluate the total response of cells to glucose load, AUCglu was calculated. AUCglu reflects the overall increase in glucose concentration after administration of glucose and tested extract (Figure 3).

In Figure 3, A is shown the glycemic curves of healthy and diabetic animals to whom *Galega officinalis* extract was administrated for 14 days. The maximum increase in blood glucose level in animals of the control group (by 282% relative to baseline level) was recorded at 20 min after administration of glucose *per os* at a dose of 1 g/kg body weight. In animals with DM, the maximum increase in blood glucose concentration (51% relative to baseline level) was recorded at 20 min after a carbohydrate load. Glucose utilization is slower in animals with diabetes compared to control animals. Maintaining a high level of glycemia in the postabsorptive period indicates a decrease in glucose tolerance in animals of this group.

Administration of *Galega officinalis* extract to control animals cause a shift of the glucose concentration peak from the 20th to the 80th minute of the oral glucose tolerance test, and an increase in the values of this indicator by 140% relative to baseline level.

In diabetic animals administered by the extract, the glycemic peak was recorded at the 60th minute of the experiment (glucose concentration increased by 92% relative to baseline level).

This nature of the curves indicates a slower absorption of glucose in the gastrointestinal tract, which leads to a more uniform load on the insular apparatus throughout the digestive process.

The use of the studied extract contributed to a probable increase in glucose tolerance. This led to a decrease in the integrated glycemic index—the area under the glycemic curve in control and diabetic animals by 25% and 47%, respectively (Figure 3B).

To substantiate the mechanisms of hypoglycemic action of the studied extract, a histological examination of pancreatic cells was performed and the functional activity of β-cells was assessed by the concentration of insulin and C-peptide in the blood plasma of rats in diabetes condition.

The development of diabetes mellitus was accompanied by damage to the incretory apparatus of the pancreas. An uneven distribution of the islets of Langerhans was pointed out, and in some parts their complete absence (Figure 4D). Morphometric analysis of the condition of the pancreatic islets showed that their number per standard unit area decreased by 3.6 times compared to the control group. There were also changes in the individual parameters of the islets of the pancreas: area, diameter, and volume (Table 1). The secretory cells of the islets of Langerhans revealed degenerative processes: vacuolation of the cytoplasm, foci of necrosis, blurred cell boundaries (Figure 4D). The number of β-cells and their secretory granules decreased sharply, as evidenced by the low intensity of staining with aldehyde-fuchsin. Nuclear hypertrophy was detected in intact cells, which were located alone or in the form of small clusters. Inflammatory changes were represented by uneven stroma edema and lymphocytic infiltration on the periphery of the islets.

As a result of histological examination of the pancreas of animals of the control group and animals that were administrated with injected with *Galega officinalis* extract (at doses of 600 and 1200 mg/kg), the preservation of the correct structure of the pancreatic lobes was observed. The islets of Langerhans are mostly of medium and large size, well defined, surrounded by a thin layer of connective tissue (Figure 4A–C). When stained with aldehyde-fuchsin, insulin-producing β-cells, which make up the main cell population, are concentrated mainly in the central part of the islet. No signs of inflammation or destruction were detected.

Administration of *Galega officinalis* extract at a dose of 600 mg/kg body weight causes a small increase in the number of islets of Langerhans in the standard cut area, their mean area, diameter, volume, and the number of β-cells compared to diabetes. Morphological changes generally were identical to such changes in the group of animals with diabetes. Signs of severe dystrophic changes persisted. The intensity of staining with aldehyde-fuchsin was reduced (Table 1, Figure 4D).

Instead, administration of the extract at a dose of 1200 mg/kg body weight showed an improvement in the histological structure of the islets of the pancreas, as evidenced by a statistically significant increase in the density of the islets of Langerhans (by 72.2%), their individual indicators—area (by 55.8%), diameter (33.7%) and volume (71%)—as well as the number of β-cells (39%), compared with animals with DM. Relatively preserved β-cells with signs of slight vacuolation of the cytoplasm and a decrease in the degree of degranulation were found (Table 1, Figure 4F). These results confirm the cytoprotective effect of the studied extract on the incretory apparatus of the pancreas.

The protective effect of *Galega officinalis* extract on the islet apparatus of the pancreas in the case of diabetes was confirmed by investigation of insulin and C-peptide levels, as they are indicators that characterize the functional state of beta cells. After a course of administration of *Galega officinalis* extract at a dose of 600 mg/kg body weight, the concentration of insulin and C-peptide did not change in animals of the control group, while in animals with DM increased (1.7 and 2.2 times, respectively) relative to diabetes level (Figure 5A,B). Insulin and C-peptide are secreted into the blood in equimolar amounts. The half-life of C-peptide in the blood is higher than the half-life of insulin, so the level of C-peptide is a more stable indicator of insulin secretion compared to the rapidly changing level of the hormone itself.

Since glucose homeostasis in diabetes can be disrupted not only by the loss of insulin-producing function and the development of β-cell dysfunction, but also by increasing the content of proinflammatory cytokines in the blood [29], a study of TNF-α content in experimental animals was conducted. Against the background of violation of the structural and functional organization of the islets of Langerhans in conditions of diabetes, an increase in the content of TNF-α in the pancreas was shown. In animals from the control group and rats with diabetes, who were administrated with *Galega officinalis* extract, a decrease in the content of TNF-α was demonstrated in the pancreas by 40 and 37%, respectively (Figure 5C).

### 3.2. In Vitro Antioxidant Effect of Non-Alkaloid Fraction of Galega Officinalis and Yacon Extracts

Our previous studies have shown that water extracts of yacon leaves and root tubers, as well as suspensions of root tubers, exhibit antioxidant properties [30,31]. Ethanol, as an extractant, has a wider range of bioactive compounds extraction compared to water, and its extractive ability is concentration-dependent. Therefore, in this work, we used a mixture of ethanol and water with an alcohol concentration of 30, 50, 70, and 96% to obtain the yacon leave extracts. This solvent has a pronounced extractive capacity for phenolic compounds, low toxicity, and antiseptic properties, which is important for the long-term storage of tinctures and extracts [32,33].

Since the release of bioactive compounds from medicinal plant raw materials mainly depended on the chemical nature of the solvent, temperature regime, as well as pH of the environment, in the first stage of the work, we have performed the selection of the process of active substances extraction from yacon leaves.

Comparing the efficiency of bioactive compounds extraction by 30, 50, 70, and 96% water–ethanol mixtures, it was found that the highest yield of the extract per 1 g of dry raw material was obtained in the case of using 30 and 50% ethyl alcohol. The extraction efficiency was improved by increasing the temperature to 40 °C and acidifying the medium to pH 2.3 (Figure 6A).

Given that the antioxidant activity of medical plant extracts is highly dependent on the content of their phenolic compounds [34,35], the total content of these components in yacon water-alcohol extracts was investigated.

The content of phenolic compounds in the tested extracts varied widely depending on the structure and concentration of the extractant. The highest total yield of phenolic compounds was obtained by using water-ethanol mixtures with an alcohol concentration of 30–50%. In the case of using water and 96% ethanol, the content of phenols in the extracts was significantly reduced (Figure 6B). The efficiency of phenolic compounds extraction with water-ethanol mixtures increased with increasing temperature up to 40 °C, which can be explained by the higher solubility and faster diffusion of the extractive substances under the action of the investigated factor. The highest PC content was found in 50% water-ethanol extract from yacon leaves. On the other hand, acidification of the medium to pH 2.3 did not increase the yield of PC in aqueous and aqueous-ethanol extracts.

Plant-derived phenolic bioactive compounds are multifunctional antioxidants that can neutralize ROS, bind variable transition metals that participate in ROS-generating Fenton’s reaction, neutralize free radicals formed during the intermediate stages of free radical oxidation. Given this, the evaluation of the antioxidant and antiradical potential of plant raw materials should be based on the use of a set of methods that reflect the various aspects of the manifestation of this activity.

The antioxidant activity was evaluated by two methods that differ by inactivated radicals. One of the most common indirect methods of assessing total antioxidant activity is the DPPH assay. DPPH• is quite selective and does not interact with flavonoids that do not contain hydroxyl groups in the B ring, as well as with aromatic acids containing only one hydroxyl group [36,37].

The results of the studies confirmed the high DPPH•-inhibitory capacity of 30, 50, and 70% of yacon extracts. Under the extraction process at 40 °C, the antiradical activity of these extracts was significantly higher compared to the extracts obtained at 20 °C. The highest antiradical activity was detected in 50% water-ethanol extract (40 °C, pH 2.3) (Figure 7A).

Analysis of the literature indicates a high degree of correlation of experimental data obtained using the ABTS method of assessing antiradical activity with phenolic compounds and inhibitory activity against ROS [37].

The results obtained with the use of ABTS- and DPPH-assays demonstrate a similar pattern—the high antiradical activity of 30, 50, and 70% of water-ethanol extracts of yacon. The highest activity was also demonstrated for 50% (20 °C), the lowest—for 96% of water-ethanol extracts (Figure 7B).

The antioxidant properties of phenolic compounds are not only determined by their antiradical action, but they are also able to interact with transition metals and thus inhibit the formation of free radicals. Phenolic compounds containing catechol and gallate structures can form stable complexes with transition metals ions and thus inhibit oxidation processes and protect cells and tissues from damage [34].

Total reduction ability was determined by the FRAP assay, based on the ability of reducing agents to reduce Fe(TPTZ)^3+^ to Fe(TPTZ)^2+^ [37]. The assay allows direct determination of low molecular weight antioxidants at the level of 10^−6^–10^−3^ mol/L. It was found that 50% water-ethanol extract obtained at a temperature of 40 °C and a temperature of 40 °C, pH 2.3 has the highest iron-reducing ability (Figure 7C).

From the obtained results, it can be concluded that the ability of the studied extracts to capture free stable radicals is determined by the content in their composition of biologically active substances of phenolic nature. The highest potential to block free radicals in in vitro model experiments possesses 50% water-ethanol extract.

In the next step, we compared the antioxidant activity of 50% water-ethanol extract of yacon and the non-alkaloid fraction of *Galega officinalis* extract (Table 2).

It was found that 50% water-ethanol extract of yacon is 8.7 times more effective in inhibiting DPPH•, and 6.3 times—ABTS•^+^ and shows 3.4 times higher iron-reducing ability, compared with the non-alkaloid fraction of *Galega officinalis* extract. The higher antioxidant activity of yacon extract is due to the high content of phenolic compounds, which is 3.2 times higher than the content of phenolic compounds in the extract of *Galega officinalis* (Table 2).

## 4. Discussion

This study evaluated the hypoglycemic and antioxidant action of *Galega officinalis* and the antioxidant properties of inulin-containing yacon culture, which is also known for its hypoglycemic action [38,39]. The pronounced antidiabetic effect of *Galega officinalis* extract on the model of streptozotocin-induced diabetes has been proved. The analysis of the obtained results gives ground to claim that the hypoglycemic effect of *Galega officinalis* extract is due to increased insulin secretion and improved structural and functional state of the islet apparatus of the pancreas, as well as increased glucose tolerance.

We found an increase in the number of islets of Langerhans on the standard cut area, their area, diameter, volume, and a number of β-cells in animals with diabetes, which was injected with *Galega officinalis* extract at a dose of 1200 mg/kg body weight, indicate the protective effect of the study extract on the incretory apparatus of the pancreas. It is known from the literature that the main ways of the destruction of beta cells, decreased synthesis, and secretion of insulin in type 1 diabetes is largely determined by the influence of ROS and TNF-α. In particular, today it is absolutely clear that in diabetes a specific organ selective destruction of beta cells takes place. Activated T-helpers produce interleukin (IL)-2 and γ-interferon, which activate macrophages, cytotoxic T-lymphocytes, and natural killer cells. These cells can exert a cytotoxic effect on β-cells of the pancreatic islets both specifically—by direct cytolysis and non-specifically—producing inflammatory mediators (ROS, cytokines, in particular IL-1, γ-interferon, TNF-α and others that detect toxic effects on β-cells) [40,41]. At in vivo and in vitro experiments, it was shown that TNF-α is one of the major cytokines that induces apoptosis of insulin-producing cells [42,43,44].

It is also known that the synthesis and secretion of insulin in chronic hyperglycemia is reduced as a result of damaging effects of ROS on key transcription factors in the pancreas-PDX-1 (*Pancreatic and duodenal homeobox 1*) and MafA (*Musculoaponeurotic fibrosarcoma oncogene homolog A*) [45,46].

It can be suggested that the cytoprotective effect of *Galega officinalis* extract on β-cells can be largely due to the ability of its components to prevent overproduction of ROS [47] and inhibit the formation of proinflammatory cytokine TNF-α, and thus reduce their cytotoxicity.

The fact that the *Galega officinalis* extract has a cytoprotective effect at a concentration of 1200 mg/kg of body weight (twice that for which the hypoglycemic effect was confirmed) may indicate the existence of other mechanisms of cytoprotective action. Since the size of β-cells in diabetic animals injected with *Galega officinalis* extract was comparable to the size of the cells in animals that did not receive extract, it can be assumed that the increase in the area occupied by β-cells is due to the ability of biologically active components of *Galega officinalis* extract not only to reduce the damaging effects of hyperglycemia but also to stimulate the division of pancreatic stem cells [48,49].

Therefore, the administration of *Galega officinalis* extract to animals with streptozotocin-induced diabetes has a pronounced antidiabetic effect, namely, reduces fasting blood glucose levels, glycosylated hemoglobin concentrations, improves glucose utilization, stimulates the secretion of insulin, and also has a cytoprotective effect on the incretory apparatus of the pancreas. It should be noted that the introduction of *Galega officinalis* extract in the control animals group did not significantly affect the studied parameters. Therefore, the analysis of the obtained results gives ground to claim that the non-alkaloid fraction of *Galega officinalis* extract shows its hypoglycemic properties only under conditions of high glucose concentration, which is characteristic of this pathology.

Analyzing the component composition of the studied extract [50], we can assume that the hypoglycemic effect is due to the presence of phytol, esters of palmitic acid, α-amyrin, and phytosterols or their synergistic action in its composition. According to the literature, phytol reduces insulin resistance, increases muscle sensitivity to insulin, inhibits gluconeogenesis, and regulates metabolic disorders that accompany diabetes by activating RXR (retinoid X receptor), which leads to increased expression of GLUT2 gene and glucokinase mRNA [51]. Studied extract contains esters of palmitic acid, which cause downregulation of glucose concentration in animals with an experimental model of diabetes [52]. The phytosterols (campesterol and stigmasterol) present in the extract can reduce the level of glycosylated hemoglobin [53].

Conducted in vitro research showed high antioxidant activity of 50% water-ethanol extract from yacon leaves, as well as a moderate antioxidant effect of the non-alkaloid fraction obtained from the aerial part of *Galega officinalis*. The antioxidant effect of the studied extracts is due to the peculiarities of the qualitative composition and quantitative content of polyphenolic compounds that are involved in the scanning of free radicals and affect the various stages of oxidative processes involving ROS.

The main active components of *Galega officinalis* extract, which determine its antioxidant properties are phytol [54] and flavonoids [55,56]. In vitro, it was shown that phytol, which is in the non-alkaloid fraction of *Galega officinalis* extract, exhibits antioxidant properties, which are due to the presence of a hydroxyl group in the molecule [54]. Flavonoids supplement the antioxidant action of the extract, as they are scavengers for electrons and free radicals, and therefore can eliminate the oxidation of biological substrates by free radical [55].

Biologically active components of the aboveground part of *Galega officinalis* suppress the inflammatory process in the pancreas, as evidenced by a decrease in TNF-α. The dominant components of the studied extract are methyl ester of linolenic acid, the content of which reaches 33.029% [50]. Linolenic acid and its esters are known for their anti-inflammatory properties [57]. From the literature, it is also known that extracts from the aboveground part of *Galega officinalis*, obtained with various extractants (dichloromethane, hexane, methanol, water) have a pronounced anti-inflammatory effect [58].

## 5. Conclusions

The combination of hypoglycemic and antioxidant effects of *Galega officinalis* and yacon indicates that these plants can serve as a source of biologically active substances of natural origin, which will effectively prevent not only hyperglycemia but also the development of oxidative stress. The pronounced anti-inflammatory properties of the biologically active components of *Galega officinalis* determine the prospects for its use in the treatment of diseases accompanied by activation of the inflammatory process and, in particular, diabetes mellitus. The obtained results indicate the prospects of using the studied plants as a basis for the creation of functional foods and antidiabetic drugs for complex therapy and the prevention of complications of diabetes.

## Figures and Tables

**Figure 1 antioxidants-10-01362-f001:**
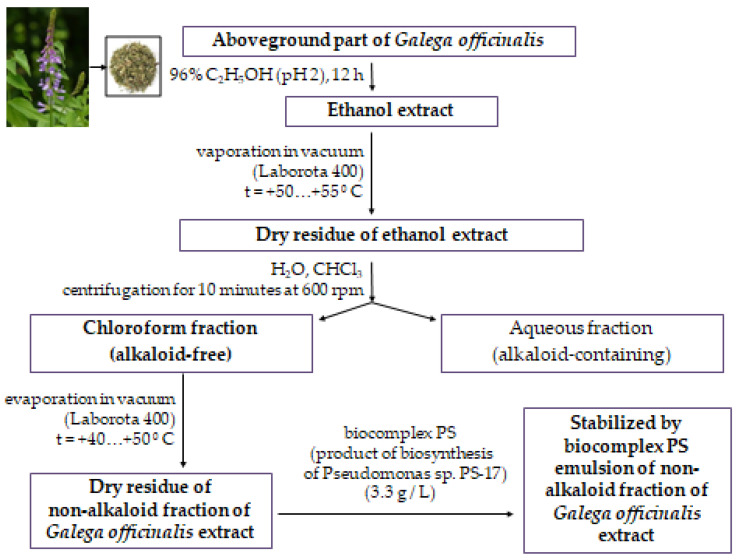
Scheme of obtaining a non-alkaloid fraction of *Galega officinalis* extract.

**Figure 2 antioxidants-10-01362-f002:**
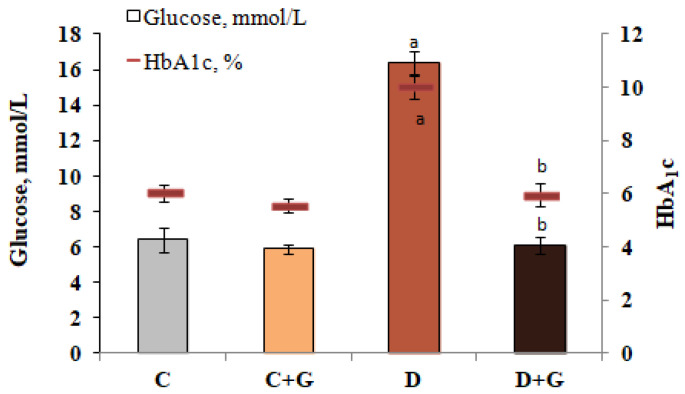
The effect of *Galega officinalis* extract on the content of glucose and glycosylated hemoglobin in normal and DM conditions (Hereinafter: control animals (C); control animals that were treated with an extract of *Galega officinalis* at a dose of 600 mg/kg per day (C + G); animals with diabetes mellitus (D); animals with diabetes that were treated with an extract of *Galega officinalis* at a dose of 600 mg/kg per day (D + G)). ^a^—*p* < 0.05, as compared with the control group; ^b^—*p* < 0.05, as compared with the diabetes mellitus group.

**Figure 3 antioxidants-10-01362-f003:**
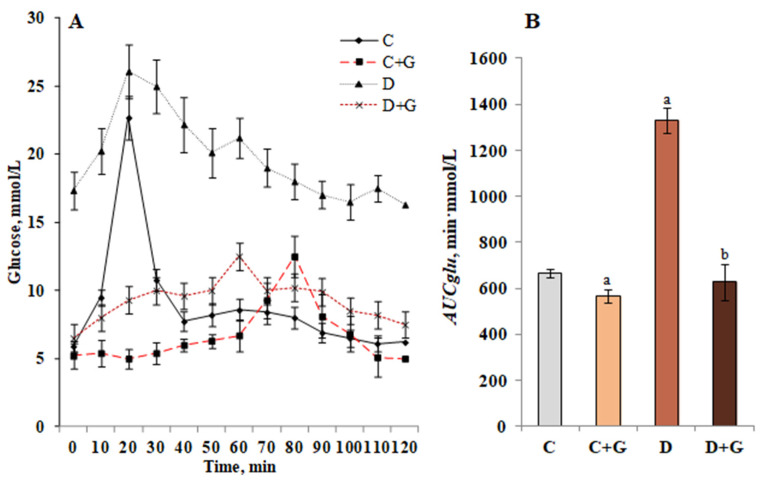
Glycemic curves (**A**) and the area under glycemic curves (**B**) in normal and DM conditions, as well as in the condition of *Galega officinalis* extract administration. ^a^—*p* < 0.05, as compared with the control group; ^b^—*p* < 0.05, as compared with the diabetes mellitus group.

**Figure 4 antioxidants-10-01362-f004:**
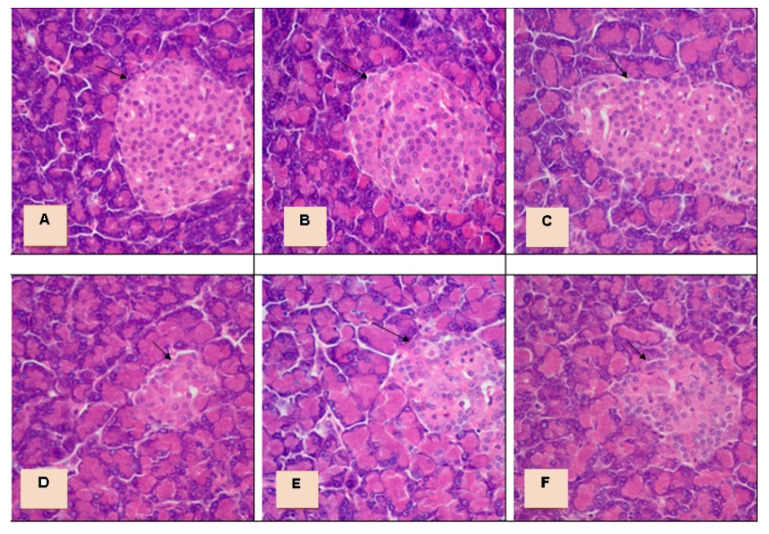
Morphological structure of the pancreas. The arrows show the islets of Langerhans. ((**A**)—control (C), (**B**)—C + G (600 mg/kg), (**C**)—C + G (1200 mg/kg)—a large number of cells in the pancreatic islets; (**D**)—diabetes (D)—small islet of Langerhans with signs of discomplexation of secretory cells with vacuolation of the cytoplasm, (**E**)—D + G (600 mg/kg)—dystrophic changes of the islet of Langerhans, (**F**)—D + G (1200 mg/kg)—increasing the number of secretory cells, reduced degree of dystrophy). Hematoxylin staining with eosin. Magnification 40 × 10.

**Figure 5 antioxidants-10-01362-f005:**
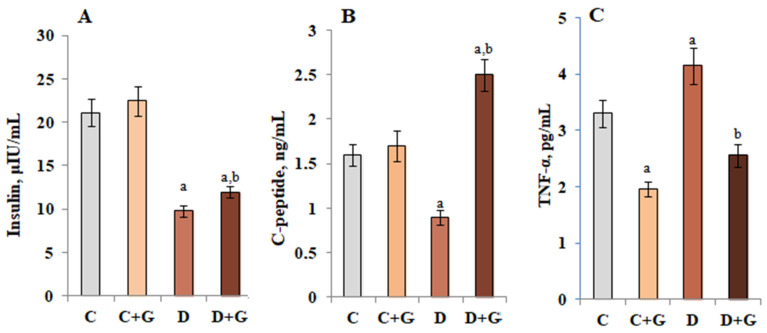
Concentrations of insulin (**A**), C-peptide (**B**) in the blood plasma and TNF-α (**C**) in the pancreas of healthy animals, under the conditions of DM and on the background of *Galega officinalis* extract administration. ^a^—*p* < 0.05, as compared with the control group; ^b^—*p* < 0.05, as compared with the diabetes mellitus group.

**Figure 6 antioxidants-10-01362-f006:**
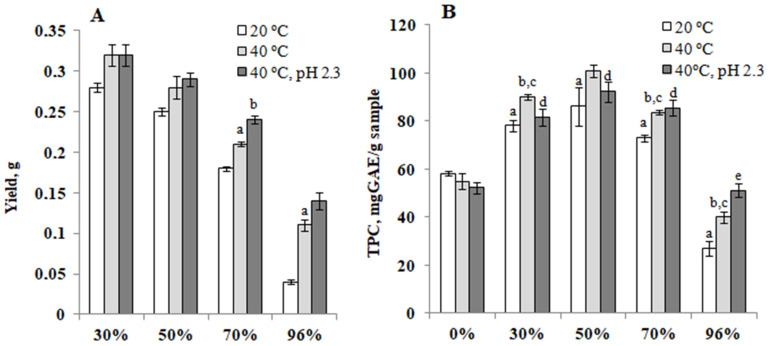
(**A**)—The extract yield (g) per 1 g of dry powder of leaves depending on the temperature and pH. ^a^—Significantly different from the water-ethanol extracts, 20 °C (*p* < 0.05); ^b^—significantly different from the water-ethanol extracts, 40 °C (*p* < 0.05). (**B**)—Content of phenolic compounds (mg GAE/g of extract) in water-ethanol extracts of yacon leaves, depending on temperature and pH. ^a^—Significantly different from the 0% ethanol, 20 °C (*p* < 0.05); ^b^—significantly different from the 0% ethanol, 40 °C (*p* < 0.05); ^c^—significantly different from the water-ethanol extracts, 20 °C (*p* < 0.05); ^d^—significantly different from the 0% ethanol, 40 °C (*p* < 0.05); ^e^—significantly different from the water-ethanol extracts, 40 °C (*p* < 0.05).

**Figure 7 antioxidants-10-01362-f007:**
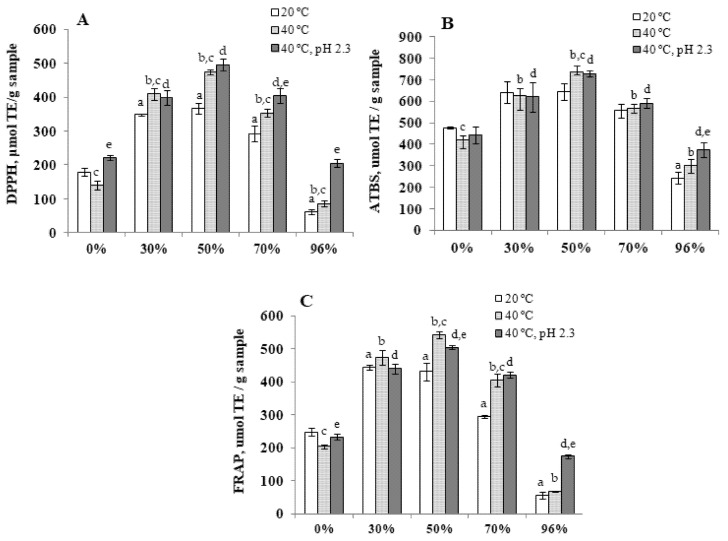
Antioxidant activity (**A**,**B**) and iron-reducing ability (**C**) of water and water-ethanol extracts of yacon leaves. ^a^—Significantly different from the 0% ethanol, 20 °C (*p* < 0.05); ^b^—significantly different from the 0% ethanol, 40 °C (*p* < 0.05); ^c^—significantly different from the water-ethanol extracts, 20 °C (*p* < 0.05); ^d^—significantly different from the 0% ethanol, 40 °C (*p* < 0.05); ^e^—significantly different from the water-ethanol extracts, 40 °C (*p* < 0.05).

**Table 1 antioxidants-10-01362-t001:** Morphometric analysis of the incretory apparatus of rats during DM and on the background of *Galega officinalis* extract administration.

Parameters				Groops		
	C	C + G_600_ *	C + G_1200_ *	D	D + G_600_ *	D + G_1200_ *
The number of islets of Langerhans on a standard area (N/10 mm^2^)	15.91 ± 0.955	14.94 ± 0.945	15.65 ± 0.595	4.420 ± 0.743 ^a^	5.673 ± 0.631 ^a^	6.640 ± 0.605 ^ab^
Islets area (μm^2^)	10,440 ± 358.5	9785 ± 523.7	10,900 ± 388.2	4620 ± 497.0 ^a^	5354 ± 387.4 ^a^	6623 ± 331.2 ^abc^
Islets diameter (μm)	160.7 ± 3.814	152.6 ± 5.658	161.9 ± 5.548	106.5 ± 4.669 ^a^	113.7 ± 3.752 ^a^	119.7 ± 4.663 ^a^
Islets volume (μm^3^)	2.20 × 10^6^ ± 1.52 × 10^5^	1.92 × 10^6^ ± 2.29 × 10^5^	2.28 × 10^6^ ± 2.38 × 10^5^	6.58 × 10^5^ ± 8.88 × 10^4 a^	7.88 × 10^5^ ± 7.40 × 10^4 a^	9.28 × 10^5^ ± 11.37 × 10^4 a^
Number of β-cells/1000 μm²	7.79 ± 0.280	7.18 ± 0.192	8.09 ± 0.308	4.74 ± 0.174 ^a^	5.12 ± 0.325 ^a^	5.62 ± 0.145 ^ab^

^a^—*p* < 0.05, as compared with the control group; ^b^—*p* < 0.05, as compared with the diabetes mellitus group; ^c^—*p* < 0.05, as compared with the group of animals D + G_600_. * C + G_600,_ C + G_1200_—control animals that were treated with extract of *Galega officinalis* at doses 600 mg and 1200 mg/kg per day, respectively; D + G_600_, D + G_1200_—animals with diabetes that were treated with extract of *Galega officinalis* at doses 600 mg and 1200 mg/kg per day, respectively.

**Table 2 antioxidants-10-01362-t002:** The content of phenolic compounds and antioxidant activity of the non-alkaloid fraction of *Galega officinalis* extract and 50% water–ethanol extract of yacon.

Sample	TPC, mg GA/g	DPPH, μmol TE/g Sample	ATBS, μmol TE/g Sample	FRAP, μmol TE/g Sample
*Galega officinalis* extract	28.77 ± 2.47	56.72 ± 1.52	114.73 ± 5.95	150.56 ± 2.76
50% water-ethanol extract of yacon	92.30 ± 4.26	495.13 ± 17.64	728.4 ± 16.6	504.5 ± 6.83

## Data Availability

Data is contained within the article.

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
