# Peer review of "Medicinal Plants Galega officinalis L. and Yacon Leaves as Potential Sources of Antidiabetic Drugs"

_antioxidants, 2021, doi:10.3390/antiox10091362_

Round 1

Reviewer 1 Report

the histograms in figures 1 and 2 report labels C, G, D but in the captions the meaning of the letters are not defined. The diagrams are therefore not easily accessible.

In figure 3, the letters indicating the different panels are positioned in between each panel. Additionally, cyrillic font are present in the picture, thus providing further confusion. 

Author Response

Response to Reviewer 1 Comments

 Point 1: the histograms in figures 1 and 2 report labels C, G, D but in the captions the meaning of the letters are not defined. The diagrams are therefore not easily accessible.

 Response 1: the marking of the letters C, G, D is given in chap. 2 Materials and Methods, subdivision. 2.4. Experimental animals. For convenience, we give the decryption of these symbols in the captions to Figs. 1 and 2.

Point 2: In figure 3, the letters indicating the different panels are positioned in between each panel. Additionally, cyrillic font are present in the picture, thus providing further confusion. 

 Response 2: in Figure 3 there was a technical error, which was eliminated, the Cyrillic font was replaced by Latin.

Reviewer 2 Report

The research topics are " Medicinal plants Galega officinalis L. and Smallanthus sonchifolius Poepp. & Endl. - potential sources of antidiabetic drugs", which is interest research, this manuscript demonstrated the cytoprotective effect of the studied extract on pancreatic cells at a dose of 1.2 g/kg, and shown an increase in the number of islets of Langerhans in the standard cut area, their average area, diameter, volume, and a number of β-cells relative to these indicators in animals with diabetes, but some information in the manuscript is not clear. It is recommended major revision.

  1. Suggest that the title of the article should be revised as Medicinal plants Galega officinalis and yacon leaves potential sources of antidiabetic drugs
  2. The abstract Line 16-20 is not clear to read, please correct it.
  3. Please provide the IACUC number
  4. Please indicate whether the animal experiment design complies with the 3R principle.
  5. Please explain the feed information of the animal diet.
  6. Please explain why the animal groups are set to 5-8 animals per group? Why not set in the same animal numbers per group?
  7. Please explain how dose 600 mg and 1200 mg/kg per day feed to rats?
  8. Please add the configuration method of Extract of Galega officinalis.
  9. Use anti-oxidant activity (Line 224,229,235) in materials and methods, but use anti-radical activity (Line 406, 425) in the results and discussion, and the recommendations are consistent.
  10. The labeling in Figure 3 needs to be revised, and the current labeling method is easy to confuse.
  11. The statistical analysis in Table 1 suggests adding statistical comparisons between groups.

Author Response

Response to Reviewer 2 Comments

Point 1: Suggest that the title of the article should be revised as Medicinal plants Galega officinalis and yacon leaves potential sources of antidiabetic drugs

Response 1: We agree with the reviewer and have made the proposed changes to the title of the article   

Point 2: The abstract Line 16-20 is not clear to read, please correct it.

Response 2:

The hypoglycemic and antioxidant properties of biologically active substances isolated from medicinal plants – Galega officinalis L. (aboveground part) and yacon (Smallanthus sonchifolius Poepp. & Endl.) (leaves), as potential sources of antidiabetic phytomedicine, have been studied. The pronounced hypoglycemic effect of the alkaloid-free fraction of Galega officinalis extract at a dose of 0.6 g/kg in experimental diabetes mellitus (DM) has been proved.

Hypoglycemic and antioxidant properties of extracts of medicinal plants Galega officinalis L. (aboveground part) and yacon (Smallanthus sonchifolius Poepp. & Endl.) (leaves) as potential sources of biologically active substances with antidiabetic action have been studied. The pronounced hypoglycemic effect of Galega officinalis extract, devoid of alkaloids, at a dose of 0.6 g / kg in experimental diabetes mellitus (EDM) has been proven.

Point 3: Please provide the IACUC number

Response 3: The protocol was conducted in compliance with the general ethical principles of animal experiments in accordance with the “General Principles of Work on Animals”, approved by the Ist National Congress of Bioethics (Kyiv, Ukraine, 2001) and the Law of Ukraine No. 3447-IV “On Protection of Animals from Cruelty” from February 26, 2006 and conforming the guidelines from Directive 2010/63/EU of the European Parliament on the protection of animals used for scientific purposes (the Directive is firmly based on the principle of the Three Rs, to replace, reduce and refine the use of animals used for scientific purposes), as well as approved by the Ethics Committee of Ivan Franko National University of Lviv, Ukraine (protocol No. 23-07-2021 from July 19, 2021).

Point 4: Please indicate whether the animal experiment design complies with the 3R principle.

Response 4: The protocol was conducted in compliance with the general ethical principles of animal experiments in accordance with the “General Principles of Work on Animals”, approved by the Ist National Congress of Bioethics (Kyiv, Ukraine, 2001) and the Law of Ukraine No. 3447-IV “On Protection of Animals from Cruelty” from February 26, 2006 and conforming the guidelines from Directive 2010/63/EU of the European Parliament on the protection of animals used for scientific purposes (the Directive is firmly based on the principle of the Three Rs, to replace, reduce and refine the use of animals used for scientific purposes), as well as approved by the Ethics Committee of Ivan Franko National University of Lviv, Ukraine (protocol No. 23-07-2021 from July 19, 2021).

Point 5: Please explain the feed information of the animal diet.                  

Response 5: For animals feeding were used a standard full-fledged balanced granular feed for laboratory rats (Rezon-1 (recipe K 120-1), Ukraine), which provided the physiological needs of their body in vitamins, trace elements and minerals.

Point 6: Please explain why the animal groups are set to 5-8 animals per group? Why not set in the same animal numbers per group?

Response 6: To study the biological effects of Galega officinalis extract at a dose of 600 mg/kg, 4 groups of animals (8 animals each) were used: control animals (C); control animals that were treated with an extract of Galega officinalis at a dose of 600 mg / kg per day (C + G); animals with diabetes mellitus (D); animals with diabetes that were treated with an extract of Galega officinalis at a dose of 600 mg / kg per day (D + G). Additionally, 2 more groups of animals (5 animals each) were used to study the cytoprotective effect of this extract at a dose of 1200 mg / kg body weight on pancreatic cells when administered to control animals and animals with experimental diabetes mellitus.

Point 7: Please explain how dose 600 mg and 1200 mg/kg per day feed to rats?

Response 7: The non-alkaloid fraction of Galega officinalis extract was administered to rats per os using a a tube at doses 600 mg / kg (one dose per day) and 1200 mg / kg (two doses of 600 mg / kg per day) daily for 14 days.

Point 8: Please add the configuration method of Extract of Galega officinalis.

Response 8: A scheme for obtaining of a non-alkaloid fraction of Galega officinalis extract were added to manuscript.  

Point 9: Use anti-oxidant activity (Line 224,229,235) in materials and methods, but use anti-radical activity (Line 406, 425) in the results and discussion, and the recommendations are consistent.

Response 9: It was taken into account this remark of the reviewer, and throughout the text the term anti-radical activity were replaced with antioxidant activity.

Point 10: The labeling in Figure 3 needs to be revised, and the current labeling method is easy to confuse.

Response 10: The technical errors in Figure 3 have been fixed.

Point 11: The statistical analysis in Table 1 suggests adding statistical comparisons between groups.

 Response 11: Table 1 additionally was added with a statistical comparison of morphometric parameters of the incretory apparatus of the pancreas between groups D+ G (600 mg / kg) and D+ G (1200 mg / kg).

Reviewer 3 Report

This is an interesting paper with some shortcoming as listed as follows:

  • All used measurement instruments must be described much more precisely (Type, Producer, country, used software...)
  • Figs: all abbreviations used in the chart must be explained on the chart or in the title of the chart
  • In the text are used one measurement units (like mg/l) and on the charts are used mmol/L - same measurement unit must be used for the volume
  • Fig 3: replace all Cyrillic letters with Latin
  • Table 1 - in the first column are some measurement units with the mentioned parameter - in the same table are some measurement units in the first row as well?! This is confusing.
  • x-axis - figure 4 ?!
  • Why on figure 5A is missing 0%?
  • References: standardize the writing of literature and pay special attention to writing journal titles that are uneven

Sincerely,

Author Response

Response to Reviewer 3 Comments

Point 1: All used measurement instruments must be described much more precisely (Type, Producer, country, used software...)

Response 1: A more detailed description of measuring instruments was added in the text of the manuscript.

Point 2: Figs: all abbreviations used in the chart must be explained on the chart or in the title of the chart

Response 2: An explanation of the abbreviations used in charts has been added to the figures captions. 

Point 3: In the text are used one measurement units (like mg/l) and on the charts are used mmol/L - same measurement unit must be used for the volume.

Response 3: In the text in g/l were expressed only the concentration of surface-active rhamnolipid biocomplex - a product of biosynthesis of the bacterial strain Pseudomonas sp. PS-17, which was used to improve the water solubility of the non-alkaloid fraction of Galega officinalis extract (line 110). The concentration of those substances that were determined experimentally were expressed in mmol/L.

Point 4: Fig 3: replace all Cyrillic letters with Latin 

Response 4: In Figure 3, a technical error occurred, which was fixed, the Cyrillic font was replaced by Latin.

Point 5: Table 1 - in the first column are some measurement units with the mentioned parameter - in the same table are some measurement units in the first row as well?! This is confusing.

Response 5:  The first column shows the units of measurement of certain parameters, the first line - groups of animals with the dose of Galega officinalis extract, which was administered to animals. To make it clearer, the designation of the groups in the first line was replaced by: instead of C + G (0.6 g / kg) - C + G600; instead of C + G (1.2 g / kg) - C + G1200; instead of D + G (0.6 g / kg) - D + G600; instead of D + G (1.2 g / kg) - D + G1200. In addition, according to the text of the article, we have unified the representation of units expressing the concentration of Galega officinalis extract, in particular, instead of 0.6 g / kg and 1.2 g / kg - 600 mg / kg and 1200 mg / kg, respectively.

Point 6: x-axis - figure 4 ?!

Response 6: in fig. 4, as in Fig. 1 and 2B the x-axis represent groups of animals.

Point 7: Why on figure 5A is missing 0%?  

Response 7: As already mentioned in the manuscript, in our previous published works it was shown that the water leave extract as well as the extract and suspension of yacon root tubers exhibit antioxidant properties. Therefore, we aimed to investigate the antioxidant activity of water-ethanol extracts from yacon leaves depending on the content of biologically active substances of phenolic nature in their composition, which in turn depends on the extraction conditions. The results of our studies indicate that water-ethanol extracts of yacon leaves show significantly higher antioxidant activity compared to water extracts. Also, the content of compounds of phenolic nature is higher in water-ethanol extracts. Therefore, the yield of water extract of yacon per 1 g of raw material, was not calculated.

Point 8:  References: standardize the writing of literature and pay special attention to writing journal titles that are uneven.

Response 8: The list of literature was standardized, special attention was paid to the writing of uneven journal titles.

Round 2

Reviewer 1 Report

The authors should carefully check their manuscript for grammar errors as well as repetition. A good starting point could be the title of the submitted work.

Author Response

Point 1: The authors should carefully check their manuscript for grammar errors as well as repetition. A good starting point could be the title of the submitted work.

Response 1: English grammar was checked and corrected with graduate philologist.

Reviewer 2 Report

The authors corrected the manuscript according to the suggestions.

Author Response

Point 1: The authors corrected the manuscript according to the suggestions.

Response 1: We are sincerely grateful for the professional review of our manuscript.

Round 3

Reviewer 1 Report

the title has been corrected and the manuscript has been revised accordingly to reviewer suggetions.